# Computer-Aided Design and Synthesis of (Functionalized quinazoline)–(α-substituted coumarin)–arylsulfonate Conjugates against Chikungunya Virus

**DOI:** 10.3390/ijms23147646

**Published:** 2022-07-11

**Authors:** Jih Ru Hwu, Animesh Roy, Shwu-Chen Tsay, Wen-Chieh Huang, Chun-Cheng Lin, Kuo Chu Hwang, Yu-Chen Hu, Fa-Kuen Shieh, Pieter Leyssen, Johan Neyts

**Affiliations:** 1Department of Chemistry, National Tsing Hua University, Hsinchu 300044, Taiwan; roy.nthu@gmail.com (A.R.); tsay.susan@gmail.com (S.-C.T.); jordanback2001@yahoo.com.tw (W.-C.H.); cclin66@mx.nthu.edu.tw (C.-C.L.); kchwang@mx.nthu.edu.tw (K.C.H.); 2Frontier Research Center on Fundamental and Applied Sciences of Matters, National Tsing Hua University, Hsinchu 300044, Taiwan; ychu@mx.nthu.edu.tw; 3Department of Chemical Engineering, National Tsing Hua University, Hsinchu 300044, Taiwan; 4Department of Chemistry, National Central University, Jhongli City 320317, Taiwan; fshieh@ncu.edu.tw; 5Rega Institute for Medical Research, Katholieke Universiteit Leuven, Minderbroedersstraat 10, B-3000 Leuven, Belgium; pieter.leyssen@kuleuven.be (P.L.); johan.neyts@kuleuven.be (J.N.)

**Keywords:** 4-anilinoquinazoline, chikungunya virus, coumarin, molecular docking, quinazolinone, sulfonate

## Abstract

Chikungunya virus (CHIKV) has repeatedly spread via the bite of an infected mosquito and affected more than 100 countries. The disease poses threats to public health and the economy in the infected locations. Many efforts have been devoted to identifying compounds that could inhibit CHIKV. Unfortunately, successful clinical candidates have not been found yet. Computations through the simulating recognition process were performed on complexation of the nsP3 protein of CHIKV with the structures of triply conjugated drug lead candidates. The outcomes provided the aid on rational design of functionalized quinazoline-(α-substituted coumarin)-arylsulfonate compounds to inhibit CHIKV in Vero cells. The molecular docking studies showed a void space around the β carbon atom of coumarin when a substituent was attached at the α position. The formed vacancy offered a good chance for a Michael addition to take place owing to steric and electronic effects. The best conjugate containing a quinazolinone moiety exhibited potency with EC_50_ = 6.46 μM, low toxicity with CC_50_ = 59.7 μM, and the selective index (SI) = 9.24. Furthermore, the corresponding 4-anilinoquinazoline derivative improved the anti-CHIKV potency to EC_50_ = 3.84 μM, CC_50_ = 72.3 μM, and SI = 18.8. The conjugate with 4-anilinoquinazoline exhibited stronger binding affinity towards the macro domain than that with quinazolinone via hydrophobic and hydrogen bond interactions.

## 1. Introduction

A number of newly emerged infectious diseases caused by RNA viruses, such as COVID-19, bring about serious public health crisis. Chikungunya virus (CHIKV) is one of these. It is a mosquito-borne alphavirus with a positive sense, single-stranded genomic RNA that causes chikungunya fever in humans [1]. The first contagious case was found in Tanzania during an outbreak of a dengue-like illness from 1952–1953 [2]. Chikungunya-infected patients often have a stooped posture due to severe joint pains and arthralgia. By 2004, a few outbreaks of CHIKV had occurred, mainly in Africa and Asia [3]. During the last decade, CHIKV has transmitted to the regions of the Americas and Europe. By early 2021, the CHIKV infection has been identified in 114 countries and territories with more than one million patients [4].

Natural sources have been sought to identify inhibitors of CHIKV [5]. Examples include β-amyrone, harringtonine, lupenone, and so forth. Harringtonine displays the most effective results with an EC_50_ of 0.24 μM with minimal cytotoxicity [6]. On the other hand, several compounds obtained by synthetic methods are active against CHIKV. They are apigenin [7], arbidol [8], 6-azauridine [7], benzylidene(cyclopropane)carbohydrazide [1], bis(benzofuran-thiazinanone)s [9], bis(benzofuran-thiazolidinone)s [9], chloroquine [10], chrysin [7], coumarin conjugates [11,12], (5,7-dihydroxy)flavones [7], mycophenolic acid [13], prothipendyl [7], purine-aminopropanol [14], purine-β-lactam hybrids [14], ribavirin [15], silybin [7], suramin [16], etc. Among these compounds, benzylidene(cyclopropane)carbohydrazide exhibits the CHIKV-induced cytopathic effect (CPE) with an EC_50_ of 5.0 μM and SI of 14 [1]. Bis-conjugates are active against CHIKV in a CPE screening assay with an EC_50_ of ~1.6–2.7 μM and SI of 133 [9]. Suramin, an existing drug with a hexasulfonate feature, inhibits CHIKV isolate replication with an EC_50_ of ~80 μM and CC_50_ >5 mM [16]. Continuous research is made by medicinal scientists to identify compounds that could effectively inhibit CHIKV. Unfortunately, numerous efforts have not yet led to a specific clinical drug [17].

Conjugated compounds are found to provide advantages with improved potency and contributions to low toxicity as antiviral agents [18]. It was found that a series of triply conjugated compounds of benzouracil-coumarin-arene exhibit anti-CHIKV activity [11]. The three units therein are connected by —SCH_2_— and —OSO_2_— joints (Figure 1). The benzouracil—SCH_2_— and arylsulfonato substituents are attached to the coumarin moiety of the conjugates **1** at the C-4 (i.e., β) and the C-7 positions, respectively. These chemically synthesized compounds show anti-CHIKV activity with an EC_50_ from 10.2–19.1 μM and SI of 11.5 [11].

Many drugs have an α,β-unsaturated enone moiety, which acts as a Michael acceptor towards enzymes and is often responsible for the resultant biological activities [19]. Examples include afatinib and osimertinib, which are among the newly developed anticancer drugs. These compounds possess a Michael addition center as the active site [19]. Attachment of a substituent at the β-position would drastically reduce the Michael addition reactivity because of the steric or electronic effect or both [20].

To improve the efficacy of triply conjugated compounds against CHIKV, we designed a new series of triple conjugates **2** possessing a quinazoline or quinazolinone moiety (Figure 1). Some quinazoline and quinazolinone derivatives show significant antimicrobial and antiviral activities [21,22]. In recent years, nitrogen-containing aromatic compounds, such as quinolines and quinazolines, have drawn the attention of scientists to their medicinal application. These small molecules show broad antiviral inhibition activities towards CHIKV [23], coronavirus (SARS-CoV-2) [24], dengue virus [25,26,27], Venezuelan equine encephalitis virus [25], etc.

Accordingly, the functionalized quinazoline and quinazolinone moieties were utilized to attach to the central coumarin nucleus at the C-3 (i.e., α) position. Simultaneously, the arylsulfonate moiety was kept at the same C-7 position. Thus, the quinazoline—SCH_2_—substituent was re-positioned from the β to the α position. Such an alternation would make the β position of conjugates **2** more available for a nucleophilic Michael addition by enzymes.

There are five structural proteins and four non-structural proteins (nsPs) in CHIKV [28]. The CHIKV nsPs perform virus RNA replication and also have many functions in host-virus interactions [29]. Owing to the enigmatic role of nsP3 in viral replication, only a few small-molecule inhibitors targeting the CHIKV nsP3 have been reported [30]. Characterization of the functional domains helps scientists to explore nsP3 as a potential drug target. The nsP3 in CHIKV has three domains [31]. Its first 160 residues (among the 530 residue-long protein) of the *N*-terminal region is the macro domain, which affects various critical processes in the alphavirus replication cycle. It includes nsP3 phosphorylation and negative strand RNA synthesis [32]. Moreover, the nsp3 macro domain plays an important role in the ADP-ribosyl-regulation of cellular proteins, ADP-ribosyl-binding, and ADP ribosyl-hydrolase activities. Hence, interaction between the macro domain and ADP-ribosylated proteins is crucial for efficient CHIKV replication. Previously, the crystal structure of the nsP3 macro domain of CHIKV has been revealed by Canard et al. [33]. The associated information makes computational approaches feasible during the development of new anti-CHIKV agents. In 2016, Seyedi and coworkers [31] reported the docking of ADP-ribose into nsP3 with a binding energy of −8.7 kcal/mol. Accordingly, we performed the molecular docking simulation studies to find out an ideal structure that would fit into the nsP3 macro domain, for which a representation of its electrostatic surface is shown in Figure 2. The skeleton of quinazoline-coumarin-arylsulfonate conjugates **2** had three nuclei and a sulfonate joint. It is analogous to the ADP-ribose, which also has three nuclei and a pyrophosphate joint [31].

Herein, we report our design and synthesis of new triple conjugates **2** as effective anti-CHIKV agents. The strategy of molecular docking in silico is undertaken to design an ideal structure for its binding with the CHIKV nsP3. As a result, the corresponding compounds with an α-substitution are conceived and synthesized to exhibit improved potency and selectivity.

## 2. Results and Discussion

### 2.1. Synthesis of Quinazolinone-(α-substituted coumarin)-sulfonate Conjugates and Structure Identification

For the synthesis of the triple conjugates with the skeleton of **2**, 7-hydroxylated coumarin **5** was considered as an ideal intermediate. It was prepared from propionic anhydride (**3**) and 2,4-dihydroxy benzaldehyde (**4**) according to the Leonetti’s procedure [34] as shown in Figure 1. Subsequent sulfonylation of hydroxycoumarin **5** was carried out by use of various benzenesulfonyl chlorides **6** and anhydrous K_2_CO_3_ between 55 and 60 °C. The desired 7-*O*-sulfonylated coumarins **7** was generated in 82–95% of yields and included different electron donating and withdrawing groups (i.e., R = *p*-OMe, *p*-F, *p*-Br, *o*-NO_2_, and *p*-NO_2_) on the arene moiety. To obtain the triple conjugates **1****0** with a (quinazolinon-2-yl)thiomethyl group at the α position of the coumarin moiety, we tried to brominate the allylic position of compounds **7**. In addition to *N*-bromosuccinimide, brominating reagents including Br_2_, PBr_3_, and Me_3_SiBr were used for the selective bromination of compound **7** at its allylic position. Reagent Br_2_ showed a lack of selectivity and led to undesired side products, mainly through β-substitution and the formation of dibromo compounds. Its use reduced the yield of desired product (<40%). Furthermore, applications of PBr_3_ and Me_3_SiBr failed to produce the desired product **8**. Finally, the reaction of coumarins **7** with *N*-bromosuccinimide [35] in anhydrous benzene at 80 °C produced the desired (bromomethyl) coumarins **8** in 78–85% of yields.

In general, CCl_4_ is an ideal solvent for allylic bromination in the Wohl-Ziegler reaction; yet, it is highly toxic to the environment. Other solvents with less toxicity than benzene were also used for the same conversion, which included toluene and CH_2_Cl_2_. Toluene underwent bromination at the benzylic position and produced benzyl bromide as a side product. This side reaction affected the yield of desired product **8**. CH_2_Cl_2_ has higher solubility than *N*-bromosuccinimide; therefore, the reaction proceeded with a faster rate and led to a decrease in the selectivity. As a result, multiple undesired by-products were formed. Ethereal solvents, such as THF and dioxane, interfered the radical reactions [36]. As a less polar solvent, benzene was found to provide better selectivity and higher yield for the formation of compound **8**.

While 2-mercapto quinazolinone **9** was coupled with allyl bromides **8** to synthesize triple conjugates **10a**–**f**, possible competition reactions were found to occur at the α,β-unsaturated lactone moiety. For instance, the application of (*N*,*N*-diisopropyl)ethylamine or 4-(dimethylamino)pyridine as a base would generate nucleophiles from **9**. These nucleophiles could react with **8** to result in ring opening of the coumarin moiety [37] or undergo a Michael addition at the β position, or both. Finally, the desired triple conjugates **10a**–**f** were obtained through *S*-alkylation of allyl bromides **8** at the C-2 thiol group of **9** exclusively, rather than at its C-4 oxygen atom. Thiolate ions of **9** are stronger nucleophiles than its oxide ions [38]. This alkylation reaction was successfully achieved with the use of K_2_CO_3_ as a base (Figure 1). Elevated temperatures (>65 °C) caused the cleavage of the sulfonate group (i.e., —SO_3_—) in the triple conjugates. To avoid the cleavage of the single bond (i.e., S-CH_2_) in **10**, alkaline solutions with a pH ranging between 10.0 and 11.5 were applied during the work-up. Finally, the new quinazolinone-coumarin-arylsulfonates **10a**–**f** were produced in very high yields (83–92%).

According to the spectroscopic data of all new compounds, their structures were identified. For example, compound **10c** had the exact mass of 511.0290 for (M + H)^+^ close to its theoretical number of 511.0293 (C_24_H_15_FN_2_O_6_S_2_ + H)^+^. There were 22 peaks detected in its ^13^C{^1^H} NMR spectrum, which are consistent with the total number of unsymmetrical carbon atoms therein. Two doublets resulting from ^1^*J*_C-F_ and ^2^*J*_C-F_ couplings resonated at 165.7 and 117.3 ppm, respectively. Two peaks in the downfield at 161.3 and 159.8 ppm were attributed to the C = O carbons in the coumarin and quinazolinone moieties. The peak at 29.3 ppm was assigned to the SCH_2_ carbon. In its ^1^H NMR spectrum, two characteristic doublets at 8.23 and 6.91 ppm corresponded to the H-8′ of quinazolinone and H-8 of coumarin protons, respectively. A distinct singlet appeared at 4.22 ppm due to the two protons in the SCH_2_ group. The carbonyl stretching frequency of compound **10c** was found in the region from 1728–1680 cm^−1^ in its IR spectrum. Furthermore, the stretching vibration of the S = O bond appeared at 1378 and 1190 cm^−1^ as the characteristics of the —SO_3_— group (for details, see Appendix A).

### 2.2. Anti-CHIKV Activity of New Triple Conjugates

For evaluating the antiviral efficacy of the newly synthesized compounds against the CHIKV Indian Ocean strain 899, we used a CPE assay on Vero cells. The materials and methods used were on the basis of the established protocol [39]. As shown in Table 1, the 50% antiviral efficacy concentration (i.e., EC_50_) is indicative of the concentration of compounds **10a**–**f** required to inhibit CHIKV replication on the host cell by 50%. The 50% cytotoxicity concentration (i.e., CC_50_) is the calculated concentration of these compounds **10a**–**f**. It reduced the measured metabolic activity of compound-treated cells by 50%. The selectivity index is a ratio (i.e., SI = CC_50_/EC_50_) that measures the therapeutic window between cytotoxicity and antiviral activity in the assay.



Among the six conjugates **1****0a**–**f**, the fluorine-containing compound **10c** exhibited the greatest anti-CHIKV activity with an EC_50_ of 6.46 μM and the best SI of 9.24 (Table 1). The triple conjugate **11** with a (quinazolinon-2-yl)thiomethyl moiety at the β-position of the coumarin is a regioisomer of **10c**, which has that thiomethyl moiety at the α-position instead. Their values of CC_50_, EC_50_, and SI are listed in Table 1 for comparison [11]. The newly synthesized α-conjugate **10c** was ~3 times more potent than the corresponding β-substituted isomer **11** against CHIKV. Meanwhile, the SI value was improved by 5.7 times in comparison with β-**11**. Improvement of the inhibitory efficiency was due to the transposition of the (quinazolinon-2-yl)thiomethyl group from the β- to the α-position of the α,β-unsaturated lactone functionality in triply conjugated compounds. The non-substituted β-position of the coumarin moiety in **10c** became readily available for a nucleophilic Michael addition to take place. These results echo the findings reported by Brummond et al. [19].

### 2.3. Design with the Aid of Molecular Docking Computations of a Conjugate with Improved Anti-CHIKV Activity

Covalent inhibition is an efficient molecular mechanism for the inhibition of enzymes and is found in the action of many drugs and biologically active natural products [40]. We applied the molecular docking method through simulating the recognition process to predict the preferable orientation of conjugates to nsP3 during their complexation [41]. Therefore, the free energy of the overall system had to be minimized by computations to find the optimized conformation of the conjugated compounds and nsP3. Understanding of the binding behavior and characters would assist us in designing an effective agent against CHIKV [42,43].

Canard et al. [33] reported that the nsP3 macro domain of CHIKV has specificity for an adenine rather than a guanine. The 4-aminopyrimidine moiety exists in structure **2b** (see Figure 1) like an adenine, while the tautomeric form of 1*H*-pyrimidin-4-one moiety exists in structure **2a** (Figure 1) like a guanine. As shown in Figure 3, the macro domain has a cavity with the size that may be ideal to accommodate a phenyl ring and have hydrophobic interaction between **2b** (R = H) and nsP3. The hydrophobicity of a compound resulting from a phenyl ring often contributes towards the increment in its antiviral activity [11]. Accordingly, we conceived an idea of developing the molecule **2b** as a better binding substrate than the molecule **2a** (R = H) for nsP3. The compound **2b** could become a more potent anti-CHIKV agent than compound **2a**.

### 2.4. Molecular Docking Studies and Protein-Substrate Interactions

Molecular docking of molecule **10c** with a CHIKV protein was initially performed at six different sites of nsP3 using the AutoDock Vina docking algorithm in the program “1-Click Docking”. Then, a total of eight docking poses of structure **2b** (R = H) were explored. Docking pose with the most negative score corresponds to the highest binding affinity. The most negative score among our computations was −10.9 kcal/mol associated with our proposed structure **2b**. This highest calculated binding affinity value was lower than −7.8 kcal/mol associated with docking of conjugate **10c** in nsP3. Meanwhile, it was also lower than −8.7 kcal/mol associated with docking of ADP-ribose in nsP3 [31]. For the in silico study, we found that conjugate **2b** was embedded into the nsP3 cavity, which is the same binding site reported for in silico studies with millions of compounds [31,44,45]. Conjugate **2b** was surrounded by 10 amino acid residues among a total of 160 present in the viral protein as shown in Figure 4. These residues were Ile11, Asn24, Arg26, Leu28, Val33, Ser110, Thr111, Tyr114, Tyr142, and Arg144, which formed a stable nsP3–**2b** complex via hydrophobic interactions and hydrogen bonds.

Results obtained from the interaction diagram for the nsP3-**2b** complex showed eight amino acid residues in the binding site involved in hydrophobic interactions; they were Ile11, Asn24, Arg26, Leu28, Val33, Tyr114, Tyr142, and Arg144 of nsP3. Interactions occurred at the different faces of 4-anilinoquinazoline, coumarin, and benzenesulfonate moieties of the triply conjugated structure **2b** with distances between 3.40 and 3.89 Å (see Table 2). It was found that four amino acid residues (i.e., Ile11, Val33, Tyr142, and Arg144 in items 1, 5, 7, and 8) involved hydrophobic interactions with the phenyl ring in the anilino group. Additionally, four intermolecular hydrogen bonds were formed between the conjugate **2b** and nsP3. Three of them came from the residues Asn24, Ser110, and Thr111, which acted as the hydrogen donors (Table 3). The —NH group of the 4-anilinoquinazoline moiety, however, acted as the hydrogen donor to interact with residue Tyr142 (item 4 in Table 3). The distances of four hydrogen bonds in the complex were calculated between 2.35 and 2.76 Å. As a consequence, Asn24 and Tyr142 participated in both the hydrophobic interaction and the H-bonding.

For the benzenesulfonyl moiety, it was bound with three hydrophobic interactions and one hydrogen bond within the protein cavity. Asn24, Arg26, and Leu28 interacted with the phenyl ring via hydrophobic interactions (Figure 5). Asn24 also formed a hydrogen bond with an oxygen atom of an —*O*-SO_2_—moiety.

Most importantly, molecular docking results indicate that a hydrogen bond was present between Ser110 and the α,β-unsaturated C = O group of coumarin in conjugate **2b**. It enabled the β-C atom of **2b** to be more electrophilic for the Michael addition to occur. These in silico data also support our hypothesis on the void space around the β-position of coumarin as shown in Figure 6. Thus, an α-substituted triple conjugate was expected to be more active than the corresponding β-substituted conjugate as an anti-CHIKV agent. A triply conjugated compound with the proposed structure **2b** would be an ideal drug lead to be investigated.

Brummond et al. [19] summarized and reported that the targeted covalent pathway has emerged as a validated approach to drug discovery. Some FDA approved drugs, such as afatinib, ibrutinib, and osimertinib, are designed to undergo an irreversible hetero-Michael addition reaction. The cysteine (Cys) residue of specific proteins often plays a vital role; a lactone moiety of a drug can inhibit the binding of Exportin-1 through covalent modification of Cys529 [46]. A crystal structure of δ-lactone covalently bound to Exportin-1 reveals that the resultant lactone is hydrolyzed to the corresponding carboxylic acid. Hydrolysis likely stabilizes the covalent adduct by decreasing the reversibility of adduct formation [47].

Coumarins, containing an α,β-unsaturated δ-lactone moiety, react with thiols to afford adducts [19]. Thiol adduct formation is one of the major pathways for the covalent inhibition of enzymes. It is also a main route for covalent inhibitor deactivation when adducts are formed with free cellular thiols, such as glutathione. Concerning the steric hindrance, α,β-unsaturated lactones with substituents (including the methyl group) at the β or the γ-position showed significant reduction in biological activities from an analog without the β-substitution. The decreased cytotoxicity is due to decreased reactivity of the substituted lactones as Michael acceptors with steric or electronic effect, or both [20,48]. These characteristics influence the ability of drugs bearing a Michael acceptor to form adducts with an enzyme containing a thiol nucleophilic center (e.g., Cys). These findings provide clues to support coumarin-containing compounds, such as conjugates **10c** and **2b,** as Michael acceptor modulators of antiviral activities. The X-ray crystallography of these drug lead-nsP3 adducts is currently under investigation.

### 2.5. Synthesis of 4-Anilinoquinazoline-coumarin-arylsulfonate Conjugate and Its Anti-CHIKV Activity

The triply conjugated compound **2b,** designed on the basis of computer-aided results, was synthesized from aniline (**12**) as shown in Figure 2. First, the 4-anilinoquinazolinone intermediate **15 [49]** was prepared from isothiocyanate **13** and 2-aminobenzonitrile (**14**). Subsequently, a number of conditions were studied to optimize the coupling reactions of the intermediate **15** and coumarin bromide **8a**. The optimal condition involving the use of K_2_CO_3_ as a base in THF at 40 °C for 5.0 h afforded the desired triple conjugate **2b** in 91% of yields. Performance of this reaction longer than 6.0 h or at elevated temperature did not increase the yield. Instead, it resulted in the cleavage of the sulfonate moiety and caused the degradation of the product **2b**.

The triple conjugate **2b** inhibited CHIKV with an EC_50_ of 3.84 μM and had an SI value of 18.8 (Table 4). With an anilino group on the quinazoline moiety, the conjugate **2b** was 2.0 times more potent than the corresponding quinazolinone conjugate **10a** with the same framework for the remaining conjugation part. The SI value of **2b** was improved by 3.68 times in comparison with **10a**. Hence, the presence of an anilinoquinazoline moiety was proved to enhance the anti-CHIKV activity. It was attributed to the hydrophobicity increased because of the additional phenyl ring and a hydrogen bond formation coming from an N-H unit. Based on our structure-activity study, we conclude that the addition of these two factors in a drug lead are advantageous to the design of some antiviral agents.

### 2.6. Lipophilicity

The molecular lipophilicity, as described by a partition coefficient (i.e., log *p*) of the new conjugated compounds **10** and **2b,** was measured by use of the “shake-flask method” [50]. Lipophilicity is an important property of drug molecules and plays a key role in drug design [51]. The log *p* value can be used to estimate the lipophilicity/hydrophobicity of a chemical entity and evaluate its pharmacokinetic profile. Consequently, we obtained the log *p* values of quinazolinone derivatives **10a**–**f** between 3.43 and 3.94 and the 4-anilinoquinazoline derivative **2b** as 5.27 (see Table 1 and Table 4).

## 3. Materials and Methods

### 3.1. General Information

Organic reactions were performed under an atmosphere of nitrogen in oven-dried glassware (~120 °C) unless otherwise noted. The R*_f_* values were measured using thin-layer chromatography (TLC) on pre-coated plates (silica gel 60 F-254). Compounds were purified using gravity column chromatography with Silicycle ultrapure silica gel (particle size 40–63 μM, 230–400 mesh). Proton nuclear magnetic resonance (^1^H NMR) spectra were recorded on a 400 MHz instrument with CDCl_3_ as an internal standard. Data are reported in parts per million (ppm) on the *δ* scale for chemical shifts and referenced from the residual protonated NMR solvent (chloroform-*d*, *δ* 7.24 ppm). Carbon-13 NMR spectra were recorded on a 100 MHz instrument and referenced from the carbon resonance of the NMR solvent at the center of the CDCl_3_ triplet (*δ* 77.0 ppm) and DMSO-*d_6_* septet (*δ* 39.5 ppm). Infrared (IR) spectra were recorded on a Fourier transform infrared (FT-IR) spectrometer. The abbreviations of absorption intensities are reported as follows: s, strong; m, medium; w, weak. High-resolution mass spectra (HRMS) were recorded on a time-of-flight (TOF) mass analyzer with electrospray ionization (ESI). The purity of all final compounds was >98.5%, as checked by HPLC (mobile phase: CH_3_CN/H_2_O, *λ* = 254 nm, column: Thermo 5 μM Hypersil ODS).

### 3.2. Preparation of Coumarin Sulfonate Intermediates ***7*** (Procedure 1)

A mixture of the sulfonyl chloride **6** (1.2 equiv) and potassium carbonate (K_2_CO_3_, 1.5 equiv) was added to a solution of 7-hydroxy-3-methylcoumarin (**5**, 1.0 equiv) in anhydrous acetone (15–17 mL). The reaction mixture was heated from 55–60 °C, stirred for between 2.0 and 2.5 h, and then cooled to 25 °C. After filtering off the inorganic solids, the filtrate was concentrated under reduced pressure to give the crude product **7**. It was then purified via silica gel column chromatography with EtOAc in hexanes as the eluent.

#### 3.2.1. 3-Methyl-2-oxo-2*H*-chromen-7-yl Benzenesulfonate (**7a**)

Procedure 1 was followed using **5** (501 mg, 2.84 mmol, 1.0 equiv), K_2_CO_3_ (589 mg, 4.26 mmol, 1.5 equiv), and benzenesulfonyl chloride (**6a**, 602 mg, 3.41 mmol, 1.2 equiv) in acetone (15 mL). The reaction mixture was stirred at 60 °C for 2.0 h and then worked up. The desired coumarin **7a** (826 mg, 2.61 mmol) was obtained in 92% of yields as white solids after purification using column chromatography (25% EtOAc in hexanes as the eluent): TLC R*_f_* 0.42 (35% EtOAc in hexanes as the eluent); mp (recrystallized from CH_2_Cl_2_) 141.4–143.2 °C; ^1^H NMR (CDCl_3_, 400 MHz) *δ* 7.83 (d, *J* = 8.0 Hz, 2 H, 2 × ArH), 7.68 (t, *J* = 8.0 Hz, 1 H, ArH), 7.55–7.51 (m, 2 H, 2 × ArH), 7.45 (s, 1 H, H-4), 7.35 (d, *J* = 8.4 Hz, 1 H, ArH), 7.00 (d, *J* = 8.4 Hz, 1 H, ArH), 6.81 (s, 1 H, ArH), 2.17 (s, 3 H, CH_3_); ^13^C NMR (CDCl_3_, 100 MHz) *δ* 160.9 (C = O), 153.2, 150.3, 138.2, 134.8, 134.6, 129.3, 128.3, 127.8, 126.1, 118.8, 118.4, 110.4, 17.1 (CH_3_); IR (KBr) 1727 (s, C = O), 1387 (m), 1321 (s, S = O), 1185 (s, S = O), 1132 (s), 1063 (s), 868 (m, S-O), 762 (m, S-O) cm^−1^; HRMS (ESI-TOF) *m*/*z* [M + H]^+^ calcd for C_16_H_12_O_5_S + H 317.0483, found 317.0481.

#### 3.2.2. 3-Methyl-2-oxo-2*H*-chromen-7-yl 4-Methoxybenzenesulfonate (**7b**)

Procedure 1 was followed using **5** (502 mg, 2.84 mmol, 1.0 equiv), K_2_CO_3_ (589 mg, 4.26 mmol, 1.5 equiv), and benzenesulfonyl chloride **6b** (704 mg, 3.41 mmol, 1.2 equiv) in acetone (15 mL). The reaction mixture was stirred at 60 °C for 2.0 h and then worked up. The desired coumarin **7b** (935 mg, 2.70 mmol) was obtained in 95% of yields as white solids after purification using column chromatography (25% EtOAc in hexanes as the eluent): TLC R*_f_* 0.41 (40% EtOAc in hexanes as the eluent); mp (recrystallized from CH_2_Cl_2_) 148.4–149.9 °C; ^1^H NMR (CDCl_3_, 400 MHz) *δ* 7.71 (d, *J* = 8.8 Hz, 2 H, 2 × ArH), 7.61 (d, *J* = 8.8 Hz, 2 H, 2 × ArH), 7.45 (s, 1 H, H-4), 7.32 (d, *J* = 8.0 Hz, 1 H, ArH), 7.08 (dd, *J* = 8.6, 2.6 Hz, 1 H, ArH), 6.88 (d, *J* = 2.4 Hz, 1 H, ArH), 3.74 (s, 3 H, OCH_3_), 2.17 (s, 3 H, CH_3_); ^13^C NMR (CDCl_3_, 100 MHz) *δ* 160.9 (C = O), 159.4, 154.1, 151.8, 148.8, 131.7, 130.0, 128.3, 125.4, 119.0, 116.0, 115.8, 111.2, 55.2 (OCH_3_), 17.3 (CH_3_); IR (KBr) 1728 (s, C = O), 1387 (m), 1321 (s, S = O), 1186 (s, S = O), 1133 (s), 1063 (s), 868 (m, S-O), 762 (m, S-O) cm^−1^; HRMS (ESI-TOF) *m*/*z* [M + H]^+^ calcd for C_17_H_14_O_6_S + H 347.0589, found 347.0584.

#### 3.2.3. 3-Methyl-2-oxo-2*H*-chromen-7-yl 4-Fluorobenzenesulfonate (**7c**)

Procedure 1 was followed using **5** (501 mg, 2.84 mmol, 1.0 equiv), K_2_CO_3_ (588 mg, 4.26 mmol, 1.5 equiv), and benzenesulfonyl chloride **6c** (663 mg, 3.41 mmol, 1.2 equiv) in acetone (16 mL). The reaction mixture was stirred at 60 °C for 2.0 h and then worked up. The desired coumarin **7c** (835 mg, 2.50 mmol) was obtained in 88% of yields as white solids after purification using column chromatography (30% EtOAc in hexanes as the eluent): TLC R*_f_* 0.42 (40% EtOAc in hexanes as the eluent); mp (recrystallized from CH_2_Cl_2_) 146.3–148.5 °C; ^1^H NMR (CDCl_3_, 400 MHz) *δ* 7.87–7.83 (m, 2 H, 2 × ArH), 7.45 (s, 1 H, H-4), 7.35 (d, *J* = 8.4 Hz, 1 H, ArH), 7.23–7.18 (m, 2 H, 2 × ArH), 6.97 (dd, *J* = 8.4, 2.4 Hz, 1 H, ArH), 6.86 (d, *J* = 2.4 Hz, 1 H, ArH), 2.18 (s, 3 H, CH_3_); ^13^C NMR (CDCl_3_, 100 MHz) *δ* 165.8 (d, *J* = 256.1 Hz, C-F), 160.9 (C = O), 153.0, 149.9, 137.9, 131.1, 131.0, 130.6, 127.8, 126.0, 118.4 (d, *J* = 7.2 Hz), 116.6 (d, *J* = 22.5 Hz), 116.5, 110.2, 17.1 (CH_3_); IR (KBr) 1727 (s, C = O), 1377 (m, S = O), 1189 (s, S = O), 1131 (m), 1069 (m), 870 (m, S-O), 742 (m, S-O), 596 (m) cm^−1^; HRMS (ESI-TOF) *m*/*z* [M + H]^+^ calcd for C_16_H_11_FO_5_S + H 335.0389, found 335.0381.

#### 3.2.4. 3-Methyl-2-oxo-2*H*-chromen-7-yl 4-Bromobenzenesulfonate (**7d**)

Procedure 1 was followed using **5** (501 mg, 2.84 mmol, 1.0 equiv), K_2_CO_3_ (589 mg, 4.26 mmol, 1.5 equiv), and benzenesulfonyl chloride **6d** (871 mg, 3.41 mmol, 1.2 equiv) in acetone (15 mL). The reaction mixture was stirred at 60 °C for 2.0 h and then worked up. The desired coumarin **7d** (1.01 g, 2.56 mmol) was obtained in 90% of yields as white solids after purification using column chromatography (25% EtOAc in hexanes as the eluent): TLC R*_f_* 0.45 (40% EtOAc in hexanes as the eluent); mp (recrystallized from CH_2_Cl_2_) 151.3–152.9 °C; ^1^H NMR (CDCl_3_, 400 MHz) *δ* 7.88 (d, *J* = 8.8 Hz, 2 H, 2 × ArH), 7.75 (d, *J* = 8.8 Hz, 2 H, 2 × ArH), 7.45 (s, 1 H, H-4), 7.35 (d, *J* = 8.4 Hz, 1 H, ArH), 6.96 (dd, *J* = 8.6, 2.2 Hz, 1 H, ArH), 6.89 (d, *J* = 2.0 Hz, 1 H, ArH), 2.17 (s, 3 H, CH_3_); ^13^C NMR (CDCl_3_, 100 MHz) *δ* 161.0 (C = O), 153.1, 149.9, 137.9, 133.6, 132.5, 129.9, 129.6, 128.1, 127.8, 126.1, 118.4, 110.3, 17.2 (CH_3_); IR (KBr) 1727 (s, C = O), 1533 (m), 1383 (m, S = O), 1192 (s, S = O), 1131 (w), 1089 (w), 871 (m, S-O), 764 (m, S-O) cm^−1^; HRMS (ESI-TOF) *m*/*z* [M + H]^+^ calcd for C_16_H_11_BrO_5_S + H 394.9588, found 394.9591.

#### 3.2.5. 3-Methyl-2-oxo-2*H*-chromen-7-yl 2-Nitrobenzenesulfonate (**7e**)

Procedure 1 was followed using **5** (498 mg, 2.84 mmol, 1.0 equiv), K_2_CO_3_ (589 mg, 4.26 mmol, 1.5 equiv), and benzenesulfonyl chloride **6e** (755 mg, 3.41 mmol, 1.2 equiv) in acetone (15 mL). The reaction mixture was stirred at 60 °C for 2.5 h and then worked up. The desired coumarin **7e** (851 mg, 2.36 mmol) was obtained in 83% of yields as pale yellow solids after purification using column chromatography (25% EtOAc in hexanes as the eluent): TLC R*_f_* 0.39 (40% EtOAc in hexanes as the eluent); mp (recrystallized from CH_2_Cl_2_) 149.4–151.2 °C; ^1^H NMR (DMSO-*d_6_*, 400 MHz) *δ* 8.20 (d, *J* = 8.0 Hz, 1 H, ArH), 8.08–8.01 (m, 2 H, 2 × ArH), 7.89–7.85 (m, 2 H, ArH + H-4), 7.67 (d, *J* = 8.4 Hz, 1 H, ArH), 7.22 (d, *J* = 2.4 Hz, 1 H, ArH), 7.10 (dd, *J* = 8.4, 2.4 Hz, 1 H, ArH), 2.07 (s, 3 H, CH_3_); ^13^C NMR (DMSO-*d_6_*, 100 MHz) *δ* 160.3 (C = O), 152.7, 148.9, 147.7, 138.3, 137.1, 133.0, 131.7, 128.9, 125.6, 125.5, 125.4, 118.9, 118.0, 109.8, 16.8 (CH_3_); IR (KBr) 1731 (s, C = O), 1532 (s, N-O), 1384 (m, S = O), 1350 (m, N-O), 1192 (s, S = O), 1133 (m), 1090 (m), 853 (m, S-O), 764 (m, S-O) cm^−1^; HRMS (ESI-TOF) *m*/*z* [M + H]^+^ calcd for C_16_H_11_NO_7_S + H 362.0334, found 362.0336.

#### 3.2.6. 3-Methyl-2-oxo-2*H*-chromen-7-yl 4-Nitrobenzenesulfonate (**7f**)

Procedure 1 was followed using **5** (499 mg, 2.84 mmol, 1.0 equiv), K_2_CO_3_ (588 mg, 4.26 mmol, 1.5 equiv), and benzenesulfonyl chloride **6f** (755 mg, 3.41 mmol, 1.2 equiv) in acetone (16 mL). The reaction mixture was stirred at 60 °C for 2.5 h and then worked up. The desired coumarin **7f** (841 mg, 2.32 mmol) was obtained in 82% of yields as pale yellow solids after purification using column chromatography (25% EtOAc in hexanes as the eluent): TLC R*_f_* 0.41 (40% EtOAc in hexanes as the eluent); mp (recrystallized from CH_2_Cl_2_) 149.7–151.8 °C; ^1^H NMR (CDCl_3_, 400 MHz) *δ* 8.38 (d, *J* = 8.8 Hz, 2 H, 2 × ArH), 8.05 (d, *J* = 8.8 Hz, 2 H, 2 × ArH), 7.46 (s, 1 H, H-4), 7.37 (d, *J* = 8.4 Hz, 1 H, ArH), 6.96 (dd, *J* = 8.4, 2.0 Hz, 1 H, ArH), 6.92 (s, 1 H, ArH), 2.18 (s, 3 H, CH_3_); ^13^C NMR (CDCl_3_, 100 MHz) *δ* 160.9 (C = O), 153.2, 150.9, 149.5, 140.3, 137.8, 129.7, 128.0, 126.5, 124.4, 118.8, 118.2, 110.2, 17.3 (CH_3_); IR (KBr) 1728 (s, C = O), 1568 (w, N-O), 1356 (m, S = O), 1337 (m, N-O), 1170 (s, S = O), 1133 (s), 1065 (m), 844 (m, S-O), 759 (m, S-O) cm^−1^; HRMS (ESI-TOF) *m*/*z* [M + H]^+^ calcd for C_16_H_11_NO_7_S + H 362.0334, found 362.0331.

### 3.3. Preparation of Bromomethyl Coumarin Intermediates ***8*** (Procedure 2)

*N*-Bromosuccinimide (NBS, 1.5 equiv) was added to a stirred solution of coumarin-sulfonate conjugate (**7**, 1.0 equiv) in anhydrous benzene (25–30 mL). The reaction mixture was refluxed at 80 °C from 16–20 h and and then cooled to 25 °C when the starting material was consumed. After filtering off the precipitated solids, the filtrate was concentrated under reduced pressure to give the crude product **8**. It was then purified via silica gel column chromatography with EtOAc in hexanes as the eluent.

#### 3.3.1. 3-Bromomethyl-2-oxo-2*H*-chromen-7-yl Benzenesulfonate (**8a**)

Procedure 2 was followed using **7a** (501 mg, 1.58 mmol, 1.0 equiv) and NBS (423 mg, 2.37 mmol, 1.5 equiv) in benzene (25 mL). The reaction mixture was refluxed for 16 h with stirring, cooled down, filtered, and concentrated under reduced pressure. The desired bromomethyl coumarin **8a** (525 mg, 1.33 mmol) was obtained in 84% of yields as white solids after purification using column chromatography (25% EtOAc in hexanes as the eluent): TLC R*_f_* 0.41 (30% EtOAc in hexanes as the eluent); mp (recrystallized from CH_2_Cl_2_) 154.2–155.8 °C; ^1^H NMR (CDCl_3_, 400 MHz) *δ* 7.84 (d, *J* = 7.6 Hz, 2 H, 2 × ArH), 7.79 (s, 1 H, H-4), 7.69 (d, *J* = 7.2 Hz, 1 H, ArH), 7.57–7.53 (m, 2 H, 2 × ArH), 7.44 (d, *J* = 8.4 Hz, 1 H, ArH), 7.05 (dd, *J* = 8.4, 2.0 Hz, 1 H, ArH), 6.87 (d, *J* = 2.0 Hz, 1 H, ArH), 4.37 (s, 2 H, CH_2_); ^13^C NMR (CDCl_3_, 100 MHz) *δ* 158.9 (C = O), 153.2, 150.3, 138.2, 134.8, 134.6, 129.3, 128.3, 127.8, 126.1, 118.8, 118.4, 110.4, 27.2 (CH_2_); IR (KBr) 1727 (s, C = O), 1387 (m), 1321 (s, S = O), 1185 (s, S = O), 1132 (s), 1063 (s), 868 (m, S-O), 762 (m, S-O) cm^−1^; HRMS (ESI-TOF) *m*/*z* [M + H]^+^ calcd for C_16_H_11_BrO_5_S + H 394.9588, found 394.9586.

#### 3.3.2. 3-Bromomethyl-2-oxo-*2H-*chromen-7-yl 4-Methoxybenzenesulfonate (**8b**)

Procedure 2 was followed using **7b** (499 mg, 1.45 mmol, 1.0 equiv) and NBS (386 mg, 2.17 mmol, 1.5 equiv) in benzene (27 mL). The reaction mixture was refluxed for 16 h with stirring, cooled down, filtered, and concentrated under reduced pressure. The desired bromomethyl coumarin **8b** (523 mg, 1.23 mmol) was obtained in 85% of yields as white solids after purification using column chromatography (25% EtOAc in hexanes as the eluent): TLC R*_f_* 0.39 (40% EtOAc in hexanes as the eluent); mp (recrystallized from CH_2_Cl_2_) 160.4–161.8 °C; ^1^H NMR (CDCl_3_, 400 MHz) *δ* 7.80 (s, 1 H, H-4), 7.72 (d, *J* = 8.8 Hz, 2 H, 2 × ArH), 7.62 (d, *J* = 8.8 Hz, 2 H, 2 × ArH), 7.32 (d, *J* = 8.0 Hz, 1 H, ArH), 7.08 (dd, *J* = 8.4, 2.4 Hz, 1 H, ArH), 6.89 (d, *J* = 2.4 Hz, 1 H, ArH), 4.38 (s, 2 H, CH_2_), 3.75 (s, 3 H, OCH_3_); ^13^C NMR (CDCl_3_, 100 MHz) *δ* 159.5 (C = O), 158.8, 154.1, 151.8, 148.9, 131.7, 130.1, 128.3, 125.5, 119.0, 116.1, 115.9, 111.3, 55.2 (OCH_3_), 27.2 (CH_2_); IR (KBr) 1727 (s, C = O), 1574 (w), 1377 (m, S = O), 1189 (m, S = O), 1131 (w), 1068 (m), 870 (m, S-O), 742 (m, S-O) cm^−1^; HRMS (ESI-TOF) *m*/*z* [M + H]^+^ calcd for C_17_H_13_BrO_6_S + H 424.9694, found 424.9687.

#### 3.3.3. 3-Bromomethyl-2-oxo-2*H*-chromen-7-yl 4-Fluorobenzenesulfonate (**8c**)

Procedure 2 was followed using **7c** (501 mg, 1.50 mmol, 1.0 equiv) and NBS (400 mg, 2.25 mmol, 1.5 equiv) in benzene (25 mL). The reaction mixture was refluxed for 18 h with stirring, cooled down, filtered, and concentrated under reduced pressure. The desired bromomethyl coumarin **8c** (501 mg, 1.21 mmol) was obtained in 81% of yields as white solids after purification using column chromatography (25% EtOAc in hexanes as the eluent): TLC R*_f_* 0.38 (40% EtOAc in hexanes as the eluent); mp (recrystallized from CH_2_Cl_2_) 159.2–161.1 °C; ^1^H NMR (CDCl_3_, 400 MHz) *δ* 7.88–7.85 (m, 2 H, 2 × ArH), 7.79 (s, 1 H, H-4), 7.45 (d, *J* = 8.4 Hz, 1 H, ArH), 7.24–7.20 (m, 2 H, 2 × ArH), 7.03 (dd, *J* = 8.8, 2.4 Hz, 1 H, ArH), 6.93 (d, *J* = 2.4 Hz, 1 H, ArH), 4.37 (s, 2 H, CH_2_); ^13^C NMR (CDCl_3_, 100 MHz) *δ* 166.0 (d, *J* = 256.5 Hz, C-F), 158.8 (C = O), 153.7, 151.2, 140.6, 138.0, 131.1, 129.0, 125.7, 119.0, 118.6, 116.8 (d, *J* = 22.7 Hz), 110.6, 27.2 (CH_2_); IR (KBr) 1727 (s, C = O), 1574 (m), 1377 (s, S = O), 1189 (s, S = O), 1131 (m), 1069 (m), 870 (m, S-O), 742 (m, S-O) cm^−1^; HRMS (ESI-TOF) *m*/*z* [M + H]^+^ calcd for C_16_H_10_FBrO_5_S + H 412.9494, found 412.9493.

#### 3.3.4. 3-Bromomethyl-2-oxo-2*H*-chromen-7-yl 4-Bromobenzenesulfonate (**8d**)

Procedure 2 was followed using **7d** (502 mg, 1.27 mmol, 1.0 equiv) and NBS (338 mg, 1.90 mmol, 1.5 equiv) in benzene (26 mL). The reaction mixture was refluxed for 18 h with stirring, cooled down, filtered, and concentrated under reduced pressure. The desired bromomethyl coumarin **8d** (492 mg, 1.04 mmol) was obtained in 82% of yields as pale yellow solids after purification using column chromatography (25% EtOAc in hexanes as the eluent: TLC R*_f_* 0.41 (40% EtOAc in hexanes as the eluent; mp (recrystallized from CH_2_Cl_2_) 168.3–169.6 °C; ^1^H NMR (CDCl_3_, 400 MHz) *δ* 7.80 (s, 1 H, H-4), 7.70 (d, *J* = 8.4 Hz, 2 H, 2 × ArH), 7.46 (d, *J* = 8.4 Hz, 2 H, 2 × ArH), 7.35 (d, *J* = 8.4 Hz, 1 H, ArH), 7.01 (dd, *J* = 8.6, 2.2 Hz, 1 H, ArH), 6.88 (d, *J* = 2.4 Hz, 1 H, ArH), 4.37 (s, 2 H, CH_2_); ^13^C NMR (CDCl_3_, 100 MHz) *δ* 158.7 (C = O), 153.7, 151.1, 140.6, 137.9, 133.5, 132.5, 129.6, 127.8, 125.6, 118.9, 118.4, 110.5, 27.2 (CH_2_); IR (KBr) 1726 (s, C = O), 1532 (s), 1349 (s, S = O), 1192 (s, S = O), 1089 (m), 872 (m, S-O), 763 (m, S-O), 619 (m) cm^−1^; HRMS (ESI-TOF) *m*/*z* [M + H]^+^ calcd for C_16_H_10_Br_2_O_5_S + H 472.8693, found 472.8689.

#### 3.3.5. 3-Bromomethyl-2-oxo-2*H*-chromen-7-yl 2-Nitrobenzenesulfonate (**8e**)

Procedure 2 was followed using **7e** (499 mg, 1.38 mmol, 1.0 equiv) and NBS (369 mg, 2.08 mmol, 1.5 equiv) in benzene (30 mL). The reaction mixture was refluxed for 20 h with stirring, cooled down, filtered, and concentrated under reduced pressure. The desired bromomethyl coumarin **8e** (488 mg, 1.11 mmol) was obtained in 80% of yields as yellow solids after purification using column chromatography (25% EtOAc in hexanes as the eluent): TLC R*_f_* 0.35 (40% EtOAc in hexanes as the eluent); mp (recrystallized from CH_2_Cl_2_) 171.4–173.1 °C; ^1^H NMR (CDCl_3_, 400 MHz) *δ* 7.99 (d, *J* = 8.0 Hz, 1 H, ArH), 7.86–7.84 (m, 2 H, 2 × ArH), 7.81 (s, 1 H, H-4), 7.72–7.70 (m, 1 H, ArH), 7.50 (d, *J* = 8.4 Hz, 1 H, ArH), 7.24–7.22 (m, 1 H, ArH), 7.15 (d, *J* = 2.0 Hz, 1 H, ArH), 4.38 (s, 2 H, CH_2_); ^13^C NMR (CDCl_3_, 100 MHz) *δ* 158.7 (C = O), 153.7, 150.6, 140.7, 138.0, 135.8, 132.1, 132.0, 129.2, 128.0, 125.7, 124.9, 118.9, 118.4, 110.5, 27.3 (CH_2_); IR (KBr) 1731 (s, C = O), 1532 (s, N-O), 1383 (s, S = O), 1349 (s, N-O), 1192 (s, S = O), 1114 (m), 979 (m), 860 (m, S-O), 764 (m, S-O) cm^−1^; HRMS (ESI-TOF) *m*/*z* [M + H]^+^ calcd for C_16_H_10_BrNO_7_S + H 439.9439, found 439.9436.

#### 3.3.6. 3-Bromomethyl-2-oxo-2*H*-chromen-7-yl 4-Nitrobenzenesulfonate (**8f**)

Procedure 2 was followed using **7f** (501 mg, 1.39 mmol, 1.0 equiv) and NBS (370 mg, 2.08 mmol, 1.5 equiv) in benzene (30 mL). The reaction mixture was refluxed for 20 h with stirring, cooled down, filtered, and concentrated under reduced pressure. The desired bromomethyl coumarin **8f** (482 mg, 1.09 mmol) was obtained in 79% of yields as yellow solids after purification using column chromatography (25% EtOAc in hexanes as the eluent): TLC R*_f_* 0.35 (40% EtOAc in hexanes as the eluent); mp (recrystallized from CH_2_Cl_2_) 173.6–175.2 °C; ^1^H NMR (CDCl_3_, 400 MHz) *δ* 7.99 (d, *J* = 8.8 Hz, 2 H, 2 × ArH), 7.82 (d, *J* = 8.8 Hz, 3 H, 2 × ArH + H-4), 7.47 (d, *J* = 8.4 Hz, 1 H, ArH), 7.03–6.98 (m, 2 H, 2 × ArH), 4.36 (s, 2 H, CH_2_); ^13^C NMR (CDCl_3_, 100 MHz) *δ* 158.7 (C = O), 153.7, 150.9, 141.5, 140.5, 138.2, 129.1, 128.8, 126.4, 125.8, 118.8, 117.8, 110.5, 27.1 (CH_2_); IR (KBr) 1729 (s, C = O), 1569 (s, N-O), 1406 (m), 1356 (s, S = O), 1339 (s, N-O), 1169 (s, S = O), 1133 (s), 844 (m, S-O), 759 (m, S-O) cm^−1^; HRMS (ESI-TOF) *m*/*z* [M + H]^+^ calcd for C_16_H_10_BrNO_7_S + H 439.9439, found 439.9440.

### 3.4. Preparation of Conjugated Compounds ***10*** and ***2b*** (Procedure 3)

A mixture of the bromomethyl coumarin **8** (1.2 equiv) and K_2_CO_3_ (1.5 equiv) was added to a stirred solution of 2-mercapto-3*H*-quinazolin-4-one (**9**, 1.0 equiv) or 4-anilinoquinazoline (**15**, 1.0 equiv) in anhydrous THF (10–12 mL). The reaction mixture was stirred at 40 °C from 5.0–6.0 h, cooled to 25 °C, and then diluted with CH_2_Cl_2_ (5.0–6.0 mL). After filtering off the inorganic solids, the filtrate was washed with an alkaline solution (i.e., NaOH/NaHPO_4_ buffer) having a pH ranging between 10.0 and 11.5. Subsequently, the organic layer was concentrated under reduced pressure to give the crude product **10** or **2b**. It was then purified via silica gel column chromatography with MeOH in CH_2_Cl_2_ as the eluent.

#### 3.4.1. 2′-Oxo-3′-([(4-oxo-3,4-dihydroquinazolin-2-yl)thio]methyl)-2*H*-chromen-7′-yl Benzenesulfonate (**10a**)

Procedure 3 was followed using **9** (50.2 mg, 0.281 mmol, 1.0 equiv), K_2_CO_3_ (58.3 mg, 0.421 mmol, 1.5 equiv), and bromomethyl coumarin **8a** (133 mg, 0.337 mmol, 1.2 equiv) in THF (5.0 mL). The reaction mixture was stirred at 40 °C for 5.0 h, cooled down, diluted with CH_2_Cl_2_, and followed by filtration. Then the filtrate was washed with NaOH/NaHPO_4_ buffer and concentrated under reduced pressure. The desired conjugated compound **10a** (127 mg, 0.256 mmol) was obtained in 92% of yields as white solids after purification using column chromatography (2.0% MeOH in CH_2_Cl_2_ as the eluent): mp (recrystallized from methanol/CH_2_Cl_2_) 211.6–212.5 °C; TLC R*_f_* 0.35 (2.0% methanol in CH_2_Cl_2_ as the eluent); ^1^H NMR (CDCl_3_, 400 MHz) *δ* 7.96 (dd, 1 H, *J* = 8.0 Hz, 1.6 Hz, ArH), 7.92 (s, 1 H, H-4′), 7.66 (dd, *J* = 8.2, 1.4 Hz, 2 H, 2 × ArH), 7.57–7.52 (m, 2 H, 2 × ArH), 7.45–7.37 (m, 3 H, 3 × ArH), 7.29–7.21 (m, 2 H, 2 × ArH), 6.80 (d, *J* = 2.0 Hz, 1 H, ArH), 6.77 (dd, *J* = 6.0, 2.0 Hz, 1 H, ArH), 4.21 (s, 2 H, SCH_2_); ^13^C NMR (DMSO-*d_6_*, 100 MHz) *δ* 161.2 (C = O), 159.7 (C = O), 154.9, 153.1, 150.3, 148.1, 140.7, 135.3, 134.6, 133.7, 129.9, 129.8, 128.3, 125.9, 125.7, 124.4, 120.0, 118.6, 118.1, 110.1, 29.3 (SCH_2_); IR (KBr) 1731 (s, C = O), 1381 (m, S = O), 1190 (s, S = O), 979 (m), 842 (m, S-O), 759 (m, S-O), 617 (w), 550 (s) cm^−1^; HRMS (ESI-TOF) *m*/*z* [M + H]^+^ calcd for C_24_H_16_N_2_O_6_S_2_ + H 493.0528, found 493.0529.

#### 3.4.2. 2′-Oxo-3′-([(4-oxo-3,4-dihydroquinazolin-2-yl)thio]methyl)-2*H*-chromen-7′-yl 4″-Methoxybenzenesulfonate (**10b**)

Procedure 3 was followed using **9** (50.4 mg, 0.281 mmol, 1.0 equiv), K_2_CO_3_ (58.2 mg, 0.421 mmol, 1.5 equiv), and bromomethyl coumarin **8b** (144 mg, 0.337 mmol, 1.2 equiv) in THF (5.0 mL). The reaction mixture was stirred at 40 °C for 5.0 h, cooled down, diluted with CH_2_Cl_2_, and followed by filtration. Then the filtrate was washed with NaOH/NaHPO_4_ buffer and concentrated under reduced pressure. The desired conjugated compound **10b** (134 mg, 0.256 mmol) was obtained in 91% of yields as white solids after purification using column chromatography (2.0% MeOH in CH_2_Cl_2_ as the eluent): mp (recrystallized from methanol/CH_2_Cl_2_) 221.7–222.8 °C; TLC R*_f_* 0.31 (2.0% methanol in CH_2_Cl_2_ as the eluent); ^1^H NMR (CDCl_3_, 400 MHz) *δ* 8.13 (d, *J* = 8.0 Hz, 1 H, ArH), 7.72–7.66 (m, 3 H, 2 × ArH + H-4′), 7.50 (d, *J* = 8.0 Hz, 2 H, 2 × ArH), 7.37 (t, *J* = 7.4 Hz, 2 H, 2 × ArH), 7.29 (d, *J* = 8.0 Hz, 2 H, 2 × ArH), 7.08 (d, *J* = 8.8 Hz, 1 H, ArH), 6.83 (s, 1 H, ArH), 4.21 (s, 2 H, SCH_2_), 3.77 (s, 3 H, OCH_3_); ^13^C NMR (DMSO-*d_6_*, 100 MHz) *δ* 163.3 (C = O), 159.6 (C = O), 154.2, 152.2, 151.7, 149.5, 148.5, 146.1, 135.3, 131.8, 130.0, 128.4, 126.8, 126.5, 126.3, 125.5, 119.8, 119.1, 117.0, 116.4, 111.2, 55.1 (OCH_3_), 29.6 (SCH_2_); IR (KBr) 2923 (m), 1726 (s, C = O), 1377 (s, S = O), 1189 (m, S = O), 840 (m, S-O), 751 (m, S-O), 612 (m), 548 (m) cm^−1^; HRMS (ESI-TOF) *m*/*z* [M-H]^+^ calcd for C_25_H_18_N_2_O_7_S_2_-H 521.0477, found 521.0466.

#### 3.4.3. 2′-Oxo-3′-([(4-oxo-3,4-dihydroquinazolin-2-yl)thio]methyl)-2*H*-chromen-7′-yl 4″-Fluorobenzenesulfonate (**10c**)

Procedure 3 was followed using **9** (50.1 mg, 0.281 mmol, 1.0 equiv), K_2_CO_3_ (58.6 mg, 0.421 mmol, 1.5 equiv), and bromomethyl coumarin **8c** (139 mg, 0.337 mmol, 1.2 equiv) in THF (6.0 mL). The reaction mixture was stirred at 40 °C for 6.0 h, cooled down, diluted with CH_2_Cl_2_, and followed by filtration. The filtrate was washed with NaOH/NaHPO_4_ buffer and concentrated under reduced pressure. The desired conjugated compound **10c** (126 mg, 0.247 mmol) was obtained in 88% of yields as white solids after purification using column chromatography (2.0% MeOH in CH_2_Cl_2_ as the eluent): mp (recrystallized from methanol/CH_2_Cl_2_) 262.4–264.2 °C; TLC R*_f_* 0.38 (2.0% methanol in CH_2_Cl_2_ as the eluent); ^1^H NMR (CDCl_3_, 400 MHz) *δ* 8.23 (d, *J* = 7.6 Hz, 1 H, ArH), 7.89–7.88 (m, 3 H, 2 × ArH + H-4′), 7.87–7.76 (m, 2 H, 2 × ArH), 7.74 (d, *J* = 8.8 Hz, 2 H, 2 × ArH), 7.58–7.45 (m, 2 H, 2 × ArH), 7.43–7.20 (m, 1 H, ArH), 7.09 (dd, *J* = 8.8, 2.0 Hz, 1 H, ArH), 6.91 (d, *J* = 2.4 Hz, 1 H, ArH), 4.22 (s, 2 H, SCH_2_); ^13^C NMR (CDCl_3_, 100 MHz) *δ* 165.7 (d, *J* = 253.8 Hz, C-F), 161.3 (C = O), 159.8 (C = O), 153.1, 150.2, 148.2, 140.8, 134.6, 131.8, 131.7, 130.1, 130.0, 129.9, 126.0, 125.8, 124.5, 120.0, 118.7, 118.2, 117.3 (d, *J* = 22.8 Hz), 110.3, 29.3 (SCH_2_); IR (KBr) 2923 (s), 1728 (s, C = O), 1378 (s, S = O), 1190 (s, S = O), 841 (m, S-O), 752 (m, S-O), 613 (m), 548 (s) cm^−1^; HRMS (ESI-TOF) *m*/*z* [M + H]^+^ calcd for C_24_H_15_FN_2_O_6_S_2_ + H 511.0293, found 511.0290.

#### 3.4.4. 2′-Oxo-3′-([(4-oxo-3,4-dihydroquinazolin-2-yl)thio]methyl)-2*H*-chromen-7′-yl 4″-Bromobenzenesulfonate (**10d**)

Procedure 3 was followed using **9** (50.7 mg, 0.281 mmol, 1.0 equiv), K_2_CO_3_ (58.3 mg, 0.421 mmol, 1.5 equiv), and bromomethyl coumarin **8d** (160 mg, 0.337 mmol, 1.2 equiv) in THF (5.0 mL). The reaction mixture was stirred at 40 °C for 5.0 h, cooled down, diluted with CH_2_Cl_2_, and followed by filtration. The filtrate was washed with NaOH/NaHPO_4_ buffer and concentrated under reduced pressure. The desired conjugated compound **10d** (144 mg, 0.253 mmol) was obtained in 90% of yields as pale yellow solids after purification using column chromatography (2.0% MeOH in CH_2_Cl_2_ as the eluent): mp (recrystallized from methanol/CH_2_Cl_2_) 282.3–284.2 °C; TLC R*_f_* 0.45 (2.0% methanol in CH_2_Cl_2_ as the eluent); ^1^H NMR (CDCl_3_, 400 MHz) *δ* 8.24 (d, *J* = 8.0 Hz, 1 H, ArH), 7.90–7.86 (m, 2 H, ArH + H-4′), 7.76–7.73 (m, 2 H, 2 × ArH), 7.57 (d, *J* = 5.2 Hz, 1 H, ArH), 7.45–7.42 (m, 3 H, 3 × ArH), 7.23–7.20 (m, 1 H, ArH), 7.10–7.07 (m, 1 H, ArH), 6.91 (d, *J* = 2.4 Hz, 1 H, ArH), 4.22 (s, 2 H, SCH_2_); ^13^C NMR (DMSO-*d*_6_, 100 MHz) *δ* 161.3 (C = O), 159.8 (C = O), 154.9, 153.2, 150.2, 148.2, 140.8, 134.6, 133.1, 133.0, 130.2, 129.9, 129.6, 126.0, 125.8, 125.7, 124.5, 120.0, 118.7, 118.3, 110.3, 29.3 (SCH_2_); IR (KBr) 2917 (w), 1727 (s, C = O), 1383 (m, S = O), 1191 (s, S = O), 853 (m, S-O), 764 (m, S-O), 622 (m), 543 (w) cm^−1^; HRMS (ESI-TOF) *m*/*z* [M-H]^+^ calcd for C_24_H_15_BrN_2_O_6_S_2_-H 568.9476, found 568.9475.

#### 3.4.5. 2′-Oxo-3′-([(4-oxo-3,4-dihydroquinazolin-2-yl)thio]methyl)-2*H*-chromen-7′-yl 2″-Nitrobenzenesulfonate (**10e**)

Procedure 3 was followed using **9** (50.3 mg, 0.281 mmol, 1.0 equiv), K_2_CO_3_ (58.7 mg, 0.421 mmol, 1.5 equiv), and bromomethyl coumarin **8e** (148 mg, 0.337 mmol, 1.2 equiv) in THF (5.0 mL). The reaction mixture was stirred at 40 °C for 6.0 h, cooled down, diluted with CH_2_Cl_2_, and followed by filtration. The filtrate was washed with NaOH/NaHPO_4_ buffer and concentrated under reduced pressure. The desired conjugated compound **10e** (128 mg, 0.239 mmol) was obtained in 85% of yields as pale yellow solids after purification using column chromatography (2.0% MeOH in CH_2_Cl_2_ as the eluent): mp (recrystallized from methanol/CH_2_Cl_2_) 272.4–273.2 °C; TLC R*_f_* 0.35 (2.0% methanol in CH_2_Cl_2_ as the eluent); ^1^H NMR (DMSO-*d_6_*, 400 MHz) *δ* 8.22 (d, *J* = 8.0 Hz, 1 H, ArH), 8.10–8.07 (m, 3 H, 2 × ArH + H-4′), 8.01 (d, *J* = 8.0 Hz, 1 H, ArH), 7.90 (t, *J* = 7.4 Hz, 1 H, ArH), 7.75 (t, *J* = 7.2 Hz, 1 H, ArH), 7.51 (d, *J* = 8.0 Hz, 2 H, 2 × ArH), 7.41 (t, *J* = 7.6 Hz, 1 H, ArH), 7.28 (d, *J* = 2.4 Hz, 1 H, ArH), 7.23 (dd, *J* = 8.8, 2.0 Hz, 1 H, ArH), 4.21 (s, 2 H, SCH_2_); ^13^C NMR (DMSO-*d*_6_, 100 MHz) *δ* 161.2 (C = O), 158.9 (C = O), 154.1, 153.8, 150.6, 150.2, 147.9, 147.9, 137.3, 134.6, 133.3, 131.8, 127.3, 126.0, 125.9, 125.8, 125.6, 125.4, 120.0, 118.1, 117.7, 116.2, 110.6, 29.3 (SCH_2_); IR (KBr) 2924 (w), 1731 (s, C = O), 1533 (s, N-O), 1383 (s, S = O), 1350 (s, N-O), 1192 (s, S = O), 853 (s, S-O), 764 (s, S-O), 627 (m), 546 (m) cm^−1^; HRMS (ESI-TOF) *m*/*z* [M + H]^+^ calcd for C_24_H_15_N_3_O_8_S_2_ + H 538.0378, found 538.0377.

#### 3.4.6. 2′-Oxo-3′-([(4-oxo-3,4-dihydroquinazolin-2-yl)thio]methyl)-2*H*-chromen-7′-yl 4″-Nitrobenzenesulfonate (**10f**)

Procedure 3 was followed using **9** (50.5 mg, 0.281 mmol, 1.0 equiv), K_2_CO_3_ (58.2 mg, 0.421 mmol, 1.5 equiv), and bromomethyl coumarin **8f** (149 mg, 0.337 mmol, 1.2 equiv) in THF (6.0 mL). The reaction mixture was stirred at 40 °C for 6.0 h, cooled down, diluted with CH_2_Cl_2_, and followed by filtration. The filtrate was washed with NaOH/NaHPO_4_ buffer and concentrated under reduced pressure. The desired conjugated compound **10f** (130 mg, 0.241 mmol) was obtained in 86% of yields as pale yellow solids after purification using column chromatography (2.0% MeOH in CH_2_Cl_2_ as the eluent): mp (recrystallized from methanol/CH_2_Cl_2_) 262.4–264.3 °C; TLC R*_f_* 0.32 (2.0% methanol in CH_2_Cl_2_ as the eluent); ^1^H NMR (DMSO-*d_6_*, 400 MHz) *δ* 8.44 (d, *J* = 8.4 Hz, 2 H, 2 × ArH), 8.21 (d, *J* = 8.4 Hz, 2 H, 2 × ArH), 8.07–8.00 (m, 2 H, ArH + H-4′), 7.75 (t, *J* = 7.6 Hz, 1 H, ArH), 7.51 (d, *J* = 8.4 Hz, 1 H, ArH), 7.43–7.40 (m, 2 H, 2 × ArH), 7.26 (s, 1 H, ArH), 7.19 (d, *J* = 8.8 Hz, 1 H, ArH), 4.24 (s, 2 H, SCH_2_); ^13^C NMR (DMSO-*d_6_*, 100 MHz) *δ* 161.2 (C = O), 159.8 (C = O), 154.2, 153.8, 151.2, 150.5, 150.3, 148.8, 137.1, 134.5, 131.1, 127.2, 126.8, 125.8, 125.1, 123.3, 120.8, 118.3, 117.6, 116.2, 110.8, 29.2 (SCH_2_); IR (KBr) 1727 (s, C = O), 1386 (m, S = O), 1320 (s, N-O), 1170 (m, S = O), 1063 (m), 868 (s, S-O), 762 (m, S-O), 625 (w), 595 (w) cm^−1^; HRMS (ESI-TOF) *m*/*z* [M + H]^+^ calcd for C_24_H_15_N_3_O_8_S_2_ + H 538.0378, found 538.0360.

#### 3.4.7. 2′-Oxo-3′-[([4-(phenylamino) quinazolin-2-yl]thio)methyl]-2*H*-chromen-7′-yl Benzenesulfonate (**2b**)

Procedure 3 was followed using quinazoline **15** (50.2 mg, 0.197 mmol, 1.0 equiv), K_2_CO_3_ (40.8 mg, 0.296 mmol, 1.5 equiv), and bromomethyl coumarin **8a** (93.8 mg, 0.237 mmol, 1.2 equiv) in THF (5.0 mL). The reaction mixture was stirred at 40 °C for 5.0 h, cooled down, diluted with CH_2_Cl_2_, and followed by filtration. The filtrate was washed with NaOH/NaHPO_4_ buffer and concentrated under reduced pressure. The desired conjugated compound **2b** (102 mg, 0.179 mmol) was obtained in 91% of yields as white solids after purification using column chromatography (1.0% MeOH in CH_2_Cl_2_ as the eluent): mp (recrystallized from methanol/CH_2_Cl_2_) 286.2–288.1 °C; TLC R*_f_* 0.45 (1.0% methanol in CH_2_Cl_2_ as the eluent); ^1^H NMR (CDCl_3_, 400 MHz) *δ* 8.43 (d, *J* = 7.6 Hz, 1 H, ArH), 7.86–7.79 (m, 3 H, 2 × ArH + H-4), 7.78–7.68 (m, 4 H, 4 × ArH), 7.66–7.57 (m, 3 H, 3 × ArH), 7.51 (t, *J* = 6.0 Hz, 1 H, ArH), 7.47–7.35 (m, 3 H, 3 × ArH), 7.14 (t, *J* = 8.0 Hz, 1 H, ArH), 7.07 (s, 1 H, ArH), 6.98 (d, *J* = 8.4 Hz, 1 H, ArH), 4.15 (s, 2 H, SCH_2_); ^13^C NMR (DMSO-*d*_6_, 100 MHz) *δ* 165.2 (N = C-S), 159.7 (C = O), 157.3, 152.8, 150.2, 150.0, 139.2, 138.5, 135.9, 133.6, 133.2, 129.8, 129.4, 128.5, 126.3, 125.6, 125.0, 124.3, 123.2, 123.1, 118.5, 118.0, 113.0, 110.6, 110.0, 29.6 (SCH_2_); IR (KBr) 3428 (w, N-H), 2929 (s), 1714 (s, C = O), 1515 (s), 1470 (m), 1384 (w, S = O), 1236 (s), 1112 (m, S = O), 819 (w, S-O), 759 (m, S-O) cm^−1^; HRMS (ESI-TOF) *m*/*z* [M + H]^+^ calcd for C_30_H_21_N_3_O_5_S_2_ + H 568.1000, found 568.1021.

### 3.5. Biological Assays

#### 3.5.1. Anti-CHIKV Activity Assay

For details, see the description in the reference [39].

#### 3.5.2. Cytotoxicity Assay

For details, also see the description in the reference [39].

### 3.6. Molecular Docking Studies and Protein-Ligand Interactions

Molecular docking of compound **2b** with CHIKV non-structural protein 3 (nsP3) was performed using the AutoDock Vina docking algorithm employed in the program “1-Click Docking” (Mcule Inc., Palo Alto, CA, USA). The X-ray crystallographic structure of nsP3 macro domains was retrieved from the Protein Data Bank (PDB ID: 3GPG). The center of binding sites was established as the default, with the Cartesian coordinates (X: 1.942, Y: 12.444, and Z: −19.292). Docking pose with the most negative docking score (−10.9 kcal/mol), corresponding to the highest binding affinity, was selected for further studies. Results obtained from the docking experiments were analyzed using the “Protein-Ligand Interaction Profiler” (provided by TU, Dresden, Germany) to explore the interactions and H-bonding in the protein-ligand complex.

## 4. Conclusions

In this study, two types of functionalized quinazoline—(α-substituted coumarin)—arylsulfonate conjugates were designed and synthesized as new anti-CHIKV leads. Their anti-CHIKV results revealed that the change of the substituent from the β- to the α-position on coumarin moiety substantially increased the inhibitory efficiency of the conjugated compounds. Increment of hydrophobicity by addition of a phenyl ring and addition of a hydrogen bond from the N-H functionality improved the anti-CHIKV potency. The most potent conjugate exerted its antiviral activity with an EC_50_ value of 3.84 μM and had an SI value of 18.8.

Molecular docking simulation was performed by complexation of the macro domain of CHIKV nsP3 with the drug lead candidates. The best conjugated structure was wrapped by different residues of nsP3 and had interactions with 10 among 160 amino acid residues. Eight hydrophobic interactions and four hydrogen bonds were observed via in silico computations. The docking results confirmed a void space around the coumarin β-position, which made the Michael addition easily able to occur. The combination of both in vitro and in silico studies provide a deep understanding of the interactions between CHIKV proteins and drug leads, which is significantly helpful in the development of new antiviral agents. Theoretical/computational studies described in this manuscript offer new insights into the understanding of experimentally obtained anti-CHIKV inhibition results in vitro. The biological activity results are in correlation as hypothesized by the molecular docking studies. A similar strategy will be of significance for medicinal chemists to design new target drug molecules.

## Data Availability

Majority of the data of this study are available within the article. The remaining is in the Appendix A.

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
