# Peer review of "Computer-Aided Design and Synthesis of (Functionalized quinazoline)–(α-substituted coumarin)–arylsulfonate Conjugates against Chikungunya Virus"

_ijms, 2022, doi:10.3390/ijms23147646_

Round 1
Reviewer 1 Report
see attached file

Author Response
For Reviewer 1:
Request 1: Nevertheless, in lines 129-131, it is mentioned that different methods were tried to obtain 8 from compound 7. Thus, detailing the other methods or conditions tested would be of interest for readers. Likewise, a brief discussion about why the method used has worked, in comparison with the others. This would allow the reading of this article to make that the conditions that have not worked are not replicated in other studies.
Response 1: In addition to N-bromosuccinimide, brominating reagents including Br2, PBr3, and Me3SiBr were used for selective bromination of compound 7 at its allylic position. Reagent Br2 showed lack of selectivity and led to undesired side products mainly through b-substitution and formation of dibromo compounds. Its use reduced the yield of desired product (<40%). Furthermore, applications of PBr3 and Me3SiBr failed to produce the desired product 8. These statements are added in lines 137-142.
Request 2: … it should be considered to use other solvents that are not benzene and are less toxic, such as toluene, for the obtaining of product 8 from compound 7?
Response 2: In general, CCl4 is an ideal solvent for allylic bromination in the Wohl–Ziegler reaction; yet it is highly toxic to the environment. Other solvents with less toxicity than benzene were also used for the same conversion, which included toluene and CH2Cl2. Toluene underwent bromination at the benzylic position and produced benzyl bromide as a side product. This side reaction affected the yield of desired product 8. CH2Cl2 has higher solubility than N-bromosuccinimide; therefore, the reaction proceeded with a faster rate and led to decrease the selectivity. As a result, multiple undesired by-products were formed. Ethereal solvents such as THF and dioxane interfered the radical reactions [36]. As a less polar solvent, benzene was found to provide better selectivity and higher yield for the formation of compound 8. The above statements are added in lines 145-154.
Request 3: In Scheme 1, compound 7 has wrong alfa and beta symbols?
Response 3: Thanks for the reviewer’s correction. Accordingly, Scheme 1 on p. 4 has been modified.
Request 4: In Scheme 1, the "n" in "n-bromo-succimide" should be a capital letter.
Response 4: In Scheme 1, the "n" in "n-bromosuccimide" is corrected to “N”.
Request 5: In line 303, it is indicated R=H for compound 2b in Scheme 2, but any "R" is shown in the compounds of Scheme 2.
Response 5: The mistake in “… compound 2b (R = H) designed on ….” is revised to “… compound 2b designed on ….” in line 322.
Request 6: In page 16, the description of Standard Procedure 3 indicates that the filtrates were washed with alkaline solutions. Please, specify which were these solutions.
Response 6: In page 16, the filtrates were washed with alkaline solution of NaOH/NaHPO4 buffer, which is added in line 563-564.
Request 7: … protocols 3.4.1-3.4.7 apply this Standard Procedure 3. However, in the descriptions, this step of washing the filtrates with these alkaline solutions is not mentioned. Please, clarify this step in procedures 3.4.1-3.4.7.
Response 7: These steps of washing the filtrates with NaOH/NaHPO4 buffer are added in procedures 3.4.1.–3.4.7. on p. 16-18.
Reviewer 2 Report
The authors report the design and synthesis of new inhibitors against Chikungunya Virus. This conjugated approach is topical and interesting. The works has been well executed overall, the level of supporting information is very good. There are a few comments to address and this manuscript will be ready for publication.
Major Comments -
1. There is no measurement of purity, this needs to be added/addressed for the final compounds
2. It would be nice to see abit more fleshing out around the 4-anilinoquinazoline scaffold. One of the lead authors has some nice background papers that should be included. This is in addition to several other papers that would put this paper in a better context in the anti-viral field.
Some of these papers that could/should be included are -
https://pubmed.ncbi.nlm.nih.gov/19223625/
https://pubmed.ncbi.nlm.nih.gov/19223625/
https://pubmed.ncbi.nlm.nih.gov/30938999/
(this paper even has a similar chemotype profile albeit in DENV)
https://pubmed.ncbi.nlm.nih.gov/32631507/
https://pubmed.ncbi.nlm.nih.gov/34217752/
https://pubmed.ncbi.nlm.nih.gov/34624490/
Minor Comments
There are several typos and English corrections
line 73 starting a sentence with 'For' is not ideal.
The double spacing between sentences is also not correct but can be dealt with a the proof stage.
Line 342-354 can be removed, this level of detail is not required.
Author Response
Request 1: There is no measurement of purity, this needs to be added/addressed for the final compounds.
Response 1: Purity measurement for the final compounds has been added in line 372-373.
Request 2: It would be nice to see a bit more fleshing out around the 4-anilinoquinazoline scaffold. One of the lead authors has some nice background papers that should be included. This is in addition to several other papers that would put this paper in a better context in the anti-viral field.
Response 2: By following the suggestions, we flesh out around the 4-anilinoquinazoline scaffold by addition of authors’ three published papers listed in Reference 18 on p. 20 (also see the text in line 66). More importantly, we thank the Reviewer for his kindness of offering us five important publish articles, which are now cited in References 23-27 on p. 20 (also see the text in lines 84-86).
Request 3: Minor Comments
There are several typos and English corrections line 73 starting a sentence with 'For' is not ideal.
The double spacing between sentences is also not correct but can be dealt with a the proof stage.
Line 342-354 can be removed, this level of detail is not required.
Response 3: Typos and English corrections have been made and went throughout the entire manuscript. In addition, the line 73 starting a sentence with 'For' is modified in lines 74-77 to “Examples include afatinib and osimertinib, which are among the newly developed anticancer drugs. These compounds possess a Michael addition centre as the active site [19].”
The double spacing between sentences is now corrected to single space.
Line 342–354 have been deleted entirely in the last paragraph of p. 11 between lines 358 and 359 accordingly.
Round 2
Reviewer 2 Report
The authors have made efforts to address all the concerns raised to a good standard. One thing is still outstanding - the HPLC traces of the final compounds need to be added to the supporting information. Once this is done the paper will be ready to publish.
Author Response
we add HPLC chromatograms for all traces of the final compounds 10a-f and 2b on p. 30 of the Supporting Information. We have revised our manuscripts based on all comments and suggestion of the reviewers and the Academic Editor. It is my wish that now IJMS accepts our manuscripts for publication.
Thank you very much!
Round 3
Reviewer 2 Report
The authors have addressed all the concerns. The HPLC traces could be abit bigger rather than thumbnails, but this can be fixed at proof stage.